# Targeted CRISPR screens reveal genes essential for *Cryptosporidium* survival in the host intestine

Lucy C. Watson [1], Katarzyna A. Sala[1], Netanya Bernitz [1], Lotta Baumgärtel [1], Mitchell A. Pallett [1], N. Bishara Marzook[1], Lorian Cobra Straker [2], Duo Peng[3], Lucy Collinson [2] & Adam Sateriale [1] ✉

The *Cryptosporidium* parasite is one of the leading causes of diarrheal morbidity and mortality in children, and adolescent infections are associated with chronic malnutrition. There are no vaccines available for protection and only one drug approved for treatment that has limited efficacy. A major barrier to developing new therapeutics is a lack of foundational knowledge of *Cryptosporidium* biology, including which parasite genes are essential for survival and virulence. Here, we iteratively improve the tools for genetically manipulating *Cryptosporidium* and develop a targeted CRISPR-based screening method to rapidly assess how the loss of individual parasite genes influence survival in vivo. Using this method, we examine the parasite's pyrimidine salvage pathway and a set of leading *Cryptosporidium* vaccine candidates. From this latter group, using inducible knockout, we determined the parasite gene known as Cp23 to be essential for survival in vivo. Parasites deficient in Cp23 were able to replicate within and emerge from infected epithelial cells, yet unable to initiate gliding motility which is required for the reinfection of neighbouring cells. The targeted screening method presented here is highly versatile and will enable researchers to more rapidly expand the knowledge base for *Cryptosporidium* infection biology, paving the way for new therapeutics.

Diarrhoeal related infections are a major cause of morbidity and mortality in children around the world[1,2]. Cryptosporidiosis is consistently found to be one of the leading causes of moderate-to-severe diarrhoeal disease in infants[3,4]. Unlike other diarrhoeal diseases that are attributed with high incidence rates, such as rotavirus or *Shigella*, there are no effective drugs or vaccines for *Cryptosporidium*. Nitazoxanide, the only Food and Drug Administration approved drug for treatment, is not effective in immunocompromised individuals and only partially effective in children, the patient population that needs intervention the most[5,6]. One reason for this scarcity of therapeutics is a historic lack of effective systems to study the parasite. While there have been recent improvements in genetic manipulation[7], drug target

identification[8–10], and animal models of infection[11], our comprehension of *Cryptosporidium* biology remains very basic. Specifically, there is a limited understanding of the *Cryptosporidium* genes that contribute to parasite fitness and survival, genes that would be the most suitable targets for therapeutic intervention.

Reverse genetic approaches have been instrumental in identifying parasite genes that influence survival and convey fitness in other Apicomplexan parasites[12–16]. *Toxoplasma*, which has historically served as a facile model for Apicomplexa, has benefited from a high transfection efficiency coupled with non-homologous end joining (NHEJ) to be at the forefront of Apicomplexa CRISPR screening[12]. Like *Cryptosporidium*, the *Plasmodium* parasite lacks NHEJ pathways, so CRISPR-Cas9

[1]Cryptosporidiosis Laboratory, The Francis Crick Institute, London, UK. [2]Electron Microscopy Science Technology Platform, The Francis Crick Institute, London, UK. [3]Chan Zuckerberg Biohub, San Francisco, CA, USA. ✉e-mail: adam.sateriale@crick.ac.uk

driven double stranded breaks can only be fixed by homologous repair. This lack of NHEJ can be leveraged to implement precise CRISPR screening, yet to date, no such screening method exists in *Cryptosporidium*. Here, we overcome the current technical barriers for *Cryptosporidium* genetic manipulation to develop a reproducible method for pooled in vivo CRISPR screens.

*Cryptosporidium* has a highly streamlined genome and lacks many basic metabolic pathways. Because of this, it is thought to be heavily reliant on its host cell for nutrients. This is particularly evident in the nucleotide salvage pathway, where *Cryptosporidium* is believed to be capable of scavenging both nucleotides and nucleotide precursors from the host enterocyte[17]. Despite a compact genome, both purine and pyrimidine salvage pathways show evidence of redundancy. The purine salvage pathway has been well studied in *Cryptosporidium*, revealing that many genes in the pathway can be independently eliminated, resulting in little or no deficit in parasite growth[17]. Surprisingly, this includes inosine monophosphate dehydrogenase (IMPDH), a gene once considered to be a crucial drug target based on metabolic mapping[18,19]. The absence of a growth defect in parasites that lack IMPDH suggests that *Cryptosporidium* can not only synthesise purine nucleotides, but also take them up directly from the infected host cell. Less is known about the pyrimidine salvage pathway in *Cryptosporidium*, however, there is still evidence of redundancy. Thymidine kinase (TK) and dihydrofolate reductase-thymidine synthase (DHFR-TS) both convert their respective substrates (deoxythymidine or deoxyuridine monophosphate) to deoxythymidine monophosphate. For this reason, TK and DHFR-TS are both independently non-essential for DNA synthesis and parasite growth[7,17]. Using our in vivo pooled CRISPR screen, we discovered that most of the pyrimidine salvage genes significantly contribute to parasite fitness within the intestine.

It is widely accepted that protective immunity can be acquired to *Cryptosporidium*, as incidences of *Cryptosporidium* infections decrease with age[3,20,21], and experimental infection with attenuated parasites leads to protection in calves and mice[11,22]. Considering the success of the vaccination campaign for rotavirus[23,24], a diarrhoeal pathogen with a similar patient population and pathogenesis, a *Cryptosporidium* vaccine is predicted to greatly reduce morbidity and mortality in children. Numerous protein antigens have been associated with protection in humans and several prominent surface proteins have been suggested as vaccine candidates[25–27]. Yet, if and how these proteins contribute to parasite fitness is unclear. Here, we assess the relative fitness contributions during infection for a panel of proposed *Cryptosporidium* vaccine candidates (collected from the literature). As Cp23 (also known as the *Cryptosporidium* immunodominant antigen 23) is one of the leading vaccine candidates, and strongly correlated with protection in humans[25], we investigated how this protein specifically contributes to parasite virulence and fitness.

## Results

### Iterative improvements to improve *Cryptosporidium* genetic manipulation

To develop CRISPR screening, we first sought to improve *Cryptosporidium* transfection efficiency. To transfect *Cryptosporidium*, dormant oocysts are treated with bile salts and warmed to body temperature to simulate the conditions of a human intestine. Under these conditions, the environmentally hardy oocysts 'excyst', releasing four motile and transfectable sporozoites that readily invade intestinal epithelial cells. To optimise transfection, we transfected *Cryptosporidium parvum* parasites with a vector that contained both luminescent (NanoLuciferase) and fluorescent (mCherry) reporters under constitutive expression and allowed the transfected parasites to infect an intestinal cell (HCT8) monolayer. Numerous electroporation programmes were trialled using transient expression, with FL115 demonstrating the highest luminescence (Supplementary Fig. 1a). Furthermore, various

combinations of bile salts and incubation media were tested for their effect on parasite transfection. Notably, incubation in sodium taurocholate led to higher transfection efficiency compared to the deoxy form, sodium taurodeoxycholate (Supplementary Fig. 1b). Combined, these adjustments resulted in more than a 50-fold improvement in transfection efficiency (Supplementary Fig. 1c, d).

As *Cryptosporidium* parasites lack NHEJ, CRISPR driven injury of the genome is required to drive homologous recombination for genetic manipulation. Although this represents a significant barrier to high-throughput screening, the parasite's lack of NHEJ very likely contributes to its uniquely high level of genetic editing specificity. Indeed, an 'off-target' genomic insertion has yet to be reported by *Cryptosporidium* researchers. Further, the parasite predominantly exists in a haploid state during its life cycle and has a minimal requirement of homology for efficient editing. To determine the smallest reliable allowance for efficient repair, we tested the efficiency of editing using differing length homology arms to integrate a nanoluciferase gene into the dispensable TK locus. A positive correlation between the length of the homology arms and the efficiency of repair was observed, with 50 bp (the longest length tested) being the most efficient, while no integration was observed in the absence of CRISPR driven injury of TK (Supplementary Fig. 1e).

To allow for pooled genetic knockout screens, we reasoned that a one-plasmid approach would be required, where a single plasmid delivers the Cas9-expression vector, a target specific gRNA, and a segment of DNA to be inserted at the target genomic locus (repair DNA). The repair DNA contains homologous flanks surrounding an expression cassette with the selectable marker and a genetic barcode for identification. Further, to enable high-throughput creation of targeted libraries, we developed a two-step Golden Gate assembly method, where first a 300 bp segment containing all gene-specific (unique) DNA is integrated into the Cas9-expression vector. In the second step of the assembly, a non-unique expression cassette is inserted that will integrate into the genome, enabling the use of virtually any combination of selection markers or reporters (Fig. 1a). To fit all unique DNA into a 300 bp oligonucleotide, we used a 50 bp gRNA which simultaneously serves as one of the homology sites for genomic integration. To test the feasibility of this approach, we generated a vector targeting TK, a gene that is known to be dispensable for parasite growth. The TK targeting vector was transfected into *Cryptosporidium parvum (C. parvum)* sporozoites that were used to infect genetically immunocompromised (interferon gamma deficient *Ifnγ⁻/⁻*) mice and luminescence was measured in mouse faecal material 7 days post transfection, comparable to what is observed when using a separate DNA repair template and a Cas9-expression vector (Fig. 1b). As we redefined the method to generate *Cryptosporidium* transgenics, we confirmed the approach's specificity via whole genome sequencing, and as expected, the only genome alteration observed was at the targeted site within the coding region of the TK gene (Fig. 1c).

### CRISPR KO screen of the pyrimidine salvage pathway

To assess the feasibility and reproducibility of a recombination-based in vivo screen, we designed a pilot screen targeting 11 genes in the parasite's pyrimidine salvage pathway (Supplementary Table 1). We ran parallel screens employing either 1 or 2 targeting vectors per gene, hence 11 or 22 gRNAs, respectively. Targeted KO vectors were transfected into *C. parvum* sporozoites, which were propagated in *Ifnγ⁻/⁻* mice under paromomycin selection (Fig. 1d). Luminescence from the integrated nanoluciferase reporter was detected in mouse faecal material 8 days post transfection, in both the 1 and 2 KO vector per gene pyrimidine salvage screens (Fig. 1e). From the output, faecal material from the infected mice, parasites were purified, their DNA extracted, and the barcodes amplified via high fidelity PCR. From the input, transfected parasites used to infect the mice, DNA was extracted and the barcodes amplified via high fidelity PCR. Both the output and

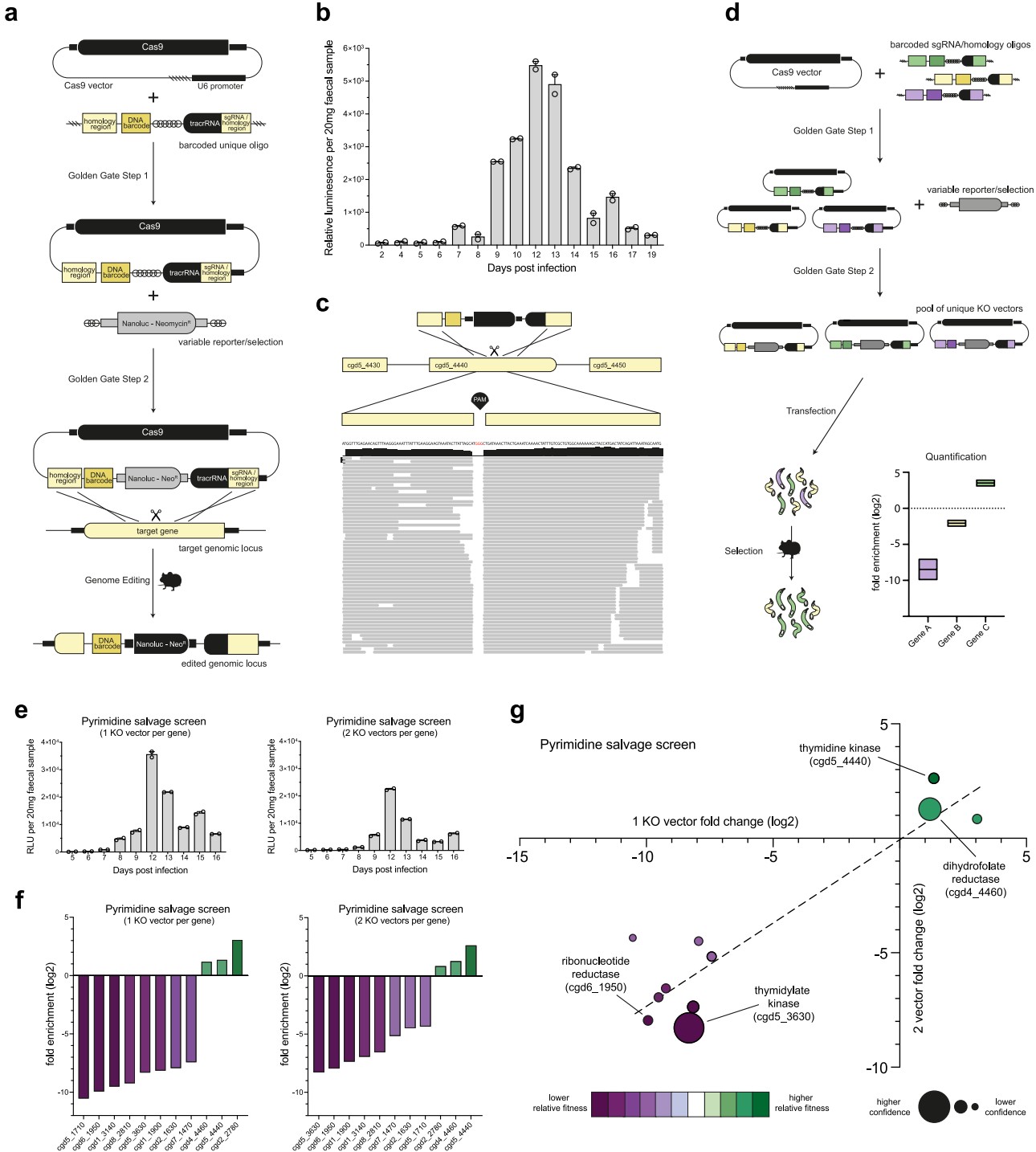

input barcodes were sequenced and used to calculate fold enrichment scores for each gene. Fold enrichment scores serve as a measure of relative fitness, genes whose barcodes show a negative enrichment are presumed to be important to parasite survival and therefore highly fitness conferring (Fig. 1f). Importantly, fold enrichment scores between the 1 and 2 KO vector screens were strongly correlated, achieving an $R^2$ of 0.842 (Fig. 1g). A separate biological repeat of the 1 KO vector screen using a separate shipment of Cryptosporidium parvum IOWAII parasites also showed good correlation, demonstrating reproducibility of the method (Supplementary Fig. 2). Of the 11 genes included in the pyrimidine salvage screen, only 3 were low fitness conferring: dihydrofolate reductase−thymidylate synthase (DHFR-TS) (cgd4_4460), thymidine kinase (TK) (cgd5_4440) and dCMP deaminase

(cgd2_2780). As mentioned previously, TK and DHFR-TS play redundant roles in synthesis of *Cryptosporidium* dTMP, making them independently non-essential for DNA synthesis and parasite growth. Thus, the results of our pilot screen effectively mirror those of previous experiments attempting individual knockouts of these genes in vivo[7,17].

## Refining diCRE-mediated genetic editing in *Cryptosporidium*

To validate our results from the pyrimidine salvage screen, we sought to use an inducible Cre-recombinase system to remove genes at their genetic locus. This method has been previously adapted for use in *C. parvum*, employing the conditional split recombinase (diCRE), which dimerises upon rapamycin addition and excises DNA between loxP recombination sites, one of which is embedded within an artificial

**Fig. 1 | In vivo CRISPR screen of the *Cryptosporidium* pyrimidine salvage pathway. a** Schematic illustrates how a knockout vector is generated. The first Golden Gate reaction occurs between a Cas9 expression plasmid and a 300 bp unique segment (containing the two 50 bp homology arms (one of which serves dual function as the gRNA), a unique DNA barcode, BsmBI restriction enzyme sites and the tracrRNA). The second Golden Gate reaction, using the BsmBI restriction enzyme sites, inserts a variable selection/reporter cassette to generate a complete knockout vector. The knockout vector then contains all the machinery to disrupt a gene of interest by inserting a variable selection cassette and barcode at the genomic locus. **b** A knockout vector targeting thymidine kinase (cgd5_4440) was generated and transfected into *C. parvum* sporozoites that were used to infect Ifny⁻/⁻ mice under paromomycin selection. Faecal samples were collected and the luminescence in faecal material was monitored. Data shows the mean faecal luminescence from a pooled cage sample (± SEM of 2 technical replicates), *n* = 4 mice. **c** Alignment of reads from whole genome sequencing of the thymidine kinase knockout strain to the *C. parvum* IOWAII genome at the site of insertion. Note the complete lack of alignment to the PAM site, which is removed by the homologous

repair event. **d** Overview of CRISPR screening method. Following construction of KO vectors (detailed in 1a), sporozoites are transfected with gene specific KO vectors and used to infect mice. Specific barcodes are then amplified via high fidelity PCR and used to calculate fold enrichment (log2[%barcode(output) / % barcodes(input)]) which measures the relative fitness contribution of each gene. **e** Mouse faecal material was collected, and luminescence was monitored in the pooled sample from the pyrimidine salvage pathway CRISPR screens at 1 and 2 KO vectors per gene. Data shows the mean faecal luminescence from a pooled cage sample (± SEM of 2 technical replicates), *n* = 4 mice. **f** Rank ordered fold enrichment scores from the 1 and 2 KO vectors per gene pyrimidine salvage CRISPR screens. The colour indicates the relative fitness contribution, with dark purple showing high fitness conferring and dark green showing low fitness conferring. **g** Comparison of the fold enrichment scores between the 1 and 2 KO vectors per gene pyrimidine salvage CRISPR screens. Confidence refers to the inverse of the 95% confidence interval when comparing the log2 fold change scores from each screen (see methods).

intron introduced into the target gene's coding sequence[28,29]. This system demonstrated high efficiency, but low-to-moderate activity in the absence of the rapamycin induction (i.e. leakiness). To refine this system, we tested a validated intron from the male gamete fusion factor HAP2 (cgd8_2220). Insertion of this intron within the nanoluciferase reporter with and without the loxP recombination site did not affect the measured luminescence during in vitro infection (Supplementary Fig. 3a). Using this validated intron/loxP combination, we generated a stable *C. parvum* diCRE parasite line targeting TK with the diCRE subunits, FRB-Cre60 and FKBP-Cre59, expressed under the same promoter, separated via a self-cleaving T2A peptide (TK-T2A-diCRE) which allows for ribosomes to 'skip' during translation and create two separate polypeptides from the same expression cassette (Supplementary Fig. 3b). However, when investigating the excision dynamics of this *C. parvum* TK-T2A-diCRE line, by infecting HCT8 monolayers in the presence or absence of rapamycin, we noted a high level of excision in the absence of rapamycin, similar to what has been reported previously (Supplementary Fig. 3c). We reasoned that the T2A skip peptide may not be functioning properly and causing background induction in *Cryptosporidium*. Consequently, we generated a stable transgenic parasite line targeting TK where the diCRE segments were under independent aldolase and tubulin promoters (TK-diCRE) (Supplementary Fig. 3d). When investigating the excision dynamics of this TK-diCRE line, we observed complete excision of the loxP-flanked TK segment at 24 h post rapamycin induction, and no measurable excision in the non-induced controls (Supplementary Fig. 3e). Using these TK-diCRE parasites (Supplementary Fig. 4a), we performed a time course to measure the dynamics of knockout at both the DNA and protein level (via a C-terminal HA tag). DNA excision had not occurred by 8 h post rapamycin treatment but was complete by 24 h, and no excision was detected in the non-induced control (Supplementary Fig. 4b). Likewise, protein levels started to decrease by 12 h post rapamycin treatment and were approximately 95% reduced compared to the non-induced controls by 24 h. In contrast, the protein levels remained high throughout the time course in the non-induced controls (Supplementary Fig. 4c, d).

### Ribonucleotide reductase is required for parasite DNA replication and survival

Ribonucleotide reductases (RNRs) catalyse the conversion of nucleotides to deoxynucleotides, an essential step for DNA synthesis in all organisms. In our pyrimidine salvage screen, RNR showed the lowest fold enrichment score, indicating a high effect on parasite fitness. Using a direct approach to disrupt the endogenous locus we were unable to derive RNR deficient parasites (Supplementary Fig. 5a), Further, our first attempt to create RNR-diCRE parasites was unsuccessful, likely due to the addition of a C-terminal HA epitope tag. It has

been suggested that the C-terminus of RNR is required for its function in other organisms and this may hold true for *Cryptosporidium*[30,31]. The second attempt to generate RNR-diCRE parasites, without the C-terminal HA epitope tag, was successful (Supplementary Fig. 4e). In vitro, rapamycin induced DNA excision of both TK- and RNR-diCRE parasites appears to follow similar dynamics, although we note some partial excision in the RNR-diCRE parasites at later timepoints (Fig. 2a, b). Further, functional deletion of both TK and RNR was confirmed using 5-ethynyl2′-deoxyuridine (EdU), a thymidine analogue. Thymidine kinase is required for EdU phosphorylation and incorporation, and it has been shown before that loss of the TK gene in *Cryptosporidium* leads to a failure to incorporate EdU[7]. In contrast, RNR deletion will not block EdU phosphorylation, but rather incorporation, as DNA synthesis cannot occur without other deoxynucleotides. We observed a complete lack of EdU incorporation in rapamycin treated TK and RNR-diCRE parasites, while EdU incorporation was observed in the non-induced and wildtype controls (Supplementary Fig. 4f). In vitro, deletion of TK led to no growth defect, whereas deletion of RNR strongly attenuated parasite growth by 24 h post infection (Fig. 2c). Similarly, rapamycin treatment of TK-diCRE infected mice lead to no growth defect, but rapamycin treatment of RNR-diCRE infected mice completely inhibited parasite growth (Fig. 2d), demonstrating that RNR is indeed essential for parasite survival, further underlining the reliability of the in vivo CRISPR screening method.

### CRISPR KO screen of *Cryptosporidium* vaccine candidates

Both the 1 and 2 KO vector per gene pyrimidine salvage screens were reproducible, but to circumvent potential low efficiency gRNAs, we chose to screen the leading *Cryptosporidium* vaccine candidates with 2 KO vectors per gene. In total, 11 vaccine candidates (22 KO vectors) were chosen due to their reported surface location and implication of immunogenicity and/or immune protection in the literature (Supplementary Table 2). Luminescence from the nanoluciferase reporter was detected in mouse faecal material around 7 days post transfection in both screens (Fig. 3a). As before, parasites were purified, DNA extracted, and barcodes were amplified via high fidelity PCR. The sequenced barcode counts were used to calculate fold enrichment scores for each gene (Fig. 3b), and importantly these scores were comparable, achieving an $R^2$ of 0.611 (Fig. 3c). Of the 11 genes in the vaccine candidate screen, only 2 were low fitness conferring: cgd6_1660 (thrombospondin repeat protein 11 (TSP11)) and cgd6_32 (apical glycoprotein 1 (AGP1)). The remaining 9 genes appeared to confer some level of fitness. Akey et al. recently demonstrated that AGP1 was dispensable for parasite survival, while apical glycoprotein 2 (AGP2) (cgd7_4330) was likely essential, both phenotypes that our pooled in vivo CRISPR screen recapitulated[32].

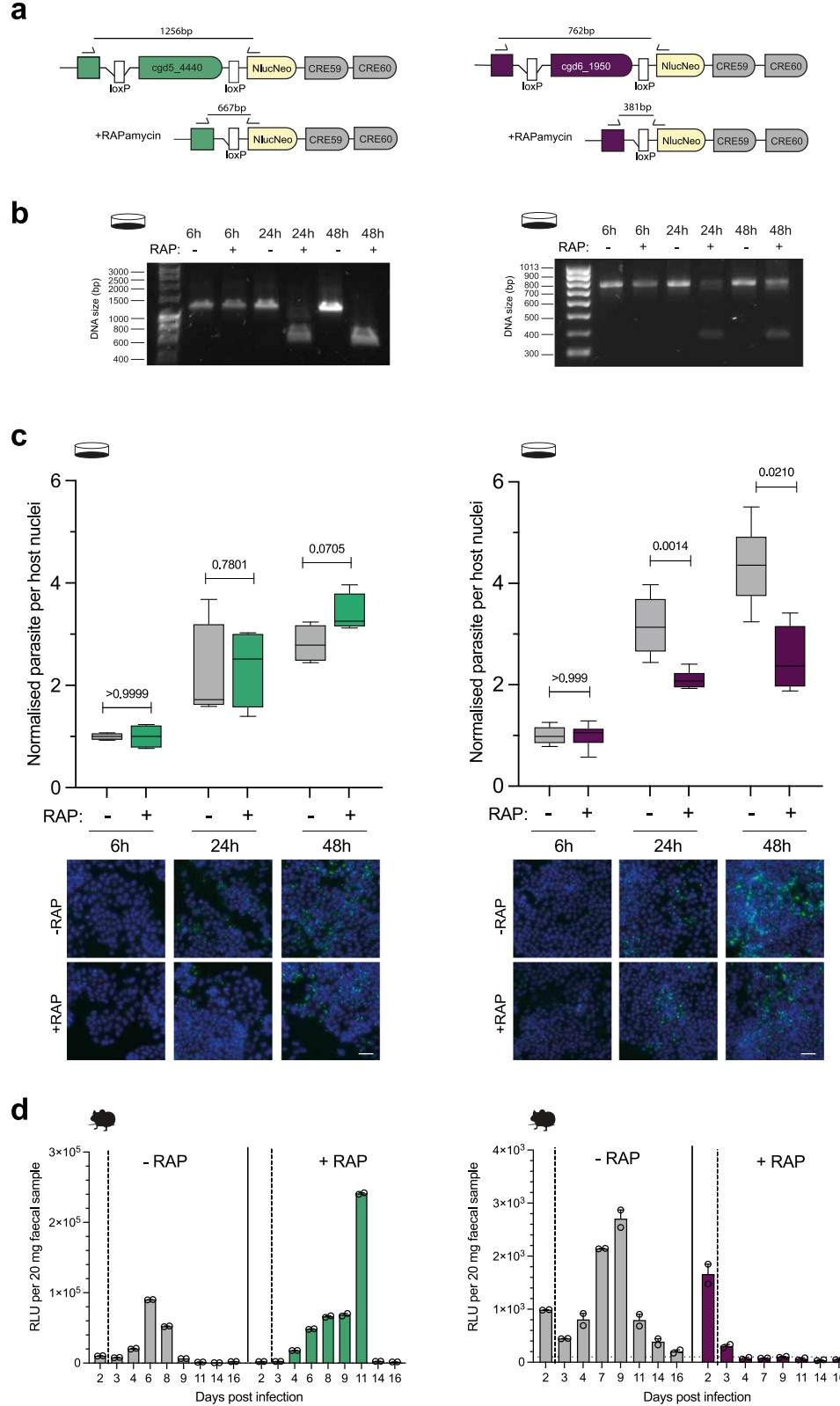

To gain more insight into growth and competition of pooled KO parasites, we conducted an additional 1 KO vector per gene vaccine candidate screen, where instead of amplifying DNA barcodes from parasites purified around the peak of infection, we amplified barcodes directly from individual faecal collections (Fig. 3d). This alternative method offered greater temporal and spatial resolution of infection where fold enrichment scores could be easily and non-invasively monitored throughout the experiment from either pooled collections or individual mice (Fig. 3e, f). Individual mice demonstrated considerable heterogeneity at day 8 of infection, which decreased over time and became more uniform by day 12. Again, this screen found that AGP1 and TSP11 were low fitness conferring relative to the other genes within the cohort.

**Fig. 2 | Validating screen results with diCRE mediated excision. a, b** TK-diCRE (shown in green) and ribonucleotide reductase (RNR) -diCRE (shown in purple) parasites were generated and used to infect an HCT8 monolayer in the presence or absence of rapamycin. Genomic DNA was extracted at 6, 24, and 48 h, and diagnostic PCRs confirmed the level of excision at the given time points. Data shown is representative of 2 biological replicates. **c** HCT8 cell monolayers infected with TK-diCRE and RNR-diCRE parasites in the presence or absence of rapamycin. At 6, 24, and 48 h the monolayers were fixed, stained, and the number of parasites per host nuclei was quantified. Box plot whiskers show the minimum and maximum values, the box shows the 25th to 75th percentile and the line shows the median.

Representative images are shown, with nuclei in blue (Hoechst), and parasites in green (Vicia villosa lectin). Scale bar = 30 μm. Data shown is representative of 2 biological replicates. Significance was determined using a two-tailed unpaired *t* test. **d** TK-diCRE or RNR-diCRE parasites were used to infect Ifnγ$^{-/-}$ mice, which were either treated with rapamycin or vehicle (DMSO) in their drinking water starting at day 2 post infection. Data shows the mean faecal luminescence from a pooled cage sample ( ± SEM of 2 technical replicates), n = 2 mice per condition. Note that a cell culture dish in the figure indicates an in vitro experiment, while a mouse silhouette indicates an in vivo experiment.

## Immunodominant antigen 23 is required for host cell invasion

Immunodominant antigen 23 (Cp23) (cgd4_3620) is one of the leading *Cryptosporidium* vaccine candidates, having been originally discovered in 1986[33]. Despite nearly 40 years of research, there is very little known about Cp23's role and function during *Cryptosporidium* infection. In our vaccine candidate screen, Cp23 displayed a low fold enrichment score, indicating a moderate-to-high effect on parasite fitness. To investigate the function of Cp23, we attempted to create a Cp23 knockout line of *Cryptosporidium* parasites but were unsuccessful (Supplementary Fig. 5). We also made an attempt to epitope tag the endogenous locus of Cp23 at the C-terminus and were unsuccessful at generating viable parasites, suggesting the epitope tag may interfere with an essential function (Supplementary Fig. 5). We were, however, successful in generating an inducible knockout Cp23-diCRE parasite line (Supplementary Fig. 6a). In vitro, the rapamycin induced DNA excision had started by 6 h and was complete by 24 h, with no excision observed in the non-induced controls (Fig. 4a, b). Loss of Cp23 was confirmed at the protein level using a commercially available monoclonal antibody. In vitro, rapamycin treatment caused a significant decrease in parasite growth after 24 h (Fig. 4c) and rapamycin treatment of Cp23-diCRE infected mice completely blocked parasite growth (Fig. 4d), further supporting that Cp23 is indeed essential for parasite survival.

Cp23 has been reported to be on the surface of the sporozoite[34], in the sporozoite trails[35], and is predicted to localise to the micronemes[36]. To localise Cp23 at high resolution, we used a commercial antibody that we genetically validated with our Cp23-diCRE parasites (Fig. 4e), coupled with expansion and super-resolution microscopy. This revealed Cp23 was expressed at the parasite's pellicle throughout the life cycle (Fig. 4f). To resolve the location further, we carried out transmission electron microscopy (TEM) with the immunogold labelled Cp23 antibody (Fig. 4g). In excysted and unexcysted sporozoites, Cp23 again appeared to primarily localise to the parasite pellicle, either at the plasma membrane or inner membrane complex (IMC). The immunogold-TEM also revealed that even in unexcysted sporozoites, Cp23 was not present in the parasite's micronemes. Further, we found no evidence that Cp23 was present in sporozoite trails (Supplementary Fig. 6b). When permeabilised with Triton X-100, sporozoites demonstrated faint staining with the Cp23 antibody (Fig. 4h). Without permeabilisation, *C. parvum* sporozoites, surprisingly, demonstrated either a complete lack of Cp23 signal or a heighted intensity of signal throughout the parasite (32.5% no signal and 67.5% heightened signal parasites) (Fig. 4h). In these high intensity parasites, we can detect the internal control antibody (CpTrpB - *Cryptosporidium* tryptophan synthase beta) at low levels. This suggests that (1) high intensity Cp23 parasites are weakly permeabilised, (2) permeabilisation with Triton X-100 affects Cp23 localisation, likely by disrupting its membrane association, and (3) Cp23 is likely not exposed on the surface of the sporozoite.

Within our synchronised in vitro model of infection, *C. parvum* parasites egress from their infected cell and re-invade nearby epithelial cells around 18 h post infection, starting another round of asexual reproduction. By 22 h, this reinvasion event is mostly complete. To determine the biological function of Cp23, we examined our Cp23-

diCRE parasites in the context of this reinfection event, when DNA excision and depletion of Cp23 has started. At 22 h, Cp23-diCRE parasites without rapamycin treatment were mostly newly invaded life stages (1n or 2n = 81% of parasites observed, n = 356/438). When Cp23 was ablated by rapamycin, fewer parasites observed were newly invaded life stages (1n or 2n = 45%, *n* = 77/173) (Fig. 4i and Supplementary Fig. 6c, e). In wildtype controls, both in the presence and absence of rapamycin, again, most of the life stages observed were newly invaded (1n or 2n = 83%, n = 176/203 and 171/213 respectively) (Fig. 4i and Supplementary Fig. 6d, e). We hypothesised that the reduced percentage of newly invaded parasites when Cp23 was ablated might be due to a defect in reinvasion. To examine this more closely, live microscopy was carried out in the presence and absence of rapamycin (Fig. 4j, k, Supplementary Movies S1–4). When Cp23 was ablated, merozoites could egress, but were then unable to move to another cell to initiate reinvasion. Apicomplexan parasites, such as *Cryptosporidium*, are known to use a method of locomotion called gliding motility, where parasites secrete proteins that are then bound by their own surface receptors, allowing for forward propulsion using an actomyosin-based complex (referred to as the glideosome)[37]. Loss of Cp23 in *Cryptosporidium* appears to specifically block this gliding motility that is essential for reinfection.

## Discussion

Reverse genetic screens play an important role in molecular biology and have transformed the field for many pathogens. Here, we overcome the current technical barriers to develop a pooled in vivo CRISPR KO method that allows for reverse genetic screening in *Cryptosporidium*. As *Cryptosporidium* lacks the molecular machinery for NHEJ, our method had to employ homologous recombination to create stable transgenics. We leveraged two important features of *Cryptosporidium* genetics: (1) the parasite spends nearly all of its life cycle in a haploid form, thus requiring only one recombination event, and (2) there is a minimal length requirement for homologous recombination, in fact 30 bp of DNA flanking the Cas9 cut site appears to be sufficient in our experiments (although efficiency increases with the size of DNA flanks). Our two-step cloning approach for generating these targeted KO vectors is designed to be versatile, allowing for an unlimited range of selection markers and reporters that can be inserted into the genome, which will allow researchers to develop new and innovative screens.

Within an infected host, *Cryptosporidium* undergoes multiple rounds of asexual reproduction before differentiating into sexual forms (male and female) that unite to allow for genetic recombination. During the sexual cycle, transgenic parasites in these CRISPR screens can mate and this has the potential to influence results. This risk is partially mitigated by two factors: (1) the parasite must complete at least three rounds of asexual replication prior to sexual differentiation and potential recombination. This gives ample time for any genes that are detrimental to parasite fitness to exert their effect; (2) *Cryptosporidium* has a relatively high rate of recombination, which should allow for genes, even those in close proximity, to segregate in a nearly random fashion[38]. In both the pyrimidine salvage and vaccine candidate screens, we identified high and low fitness conferring genes in

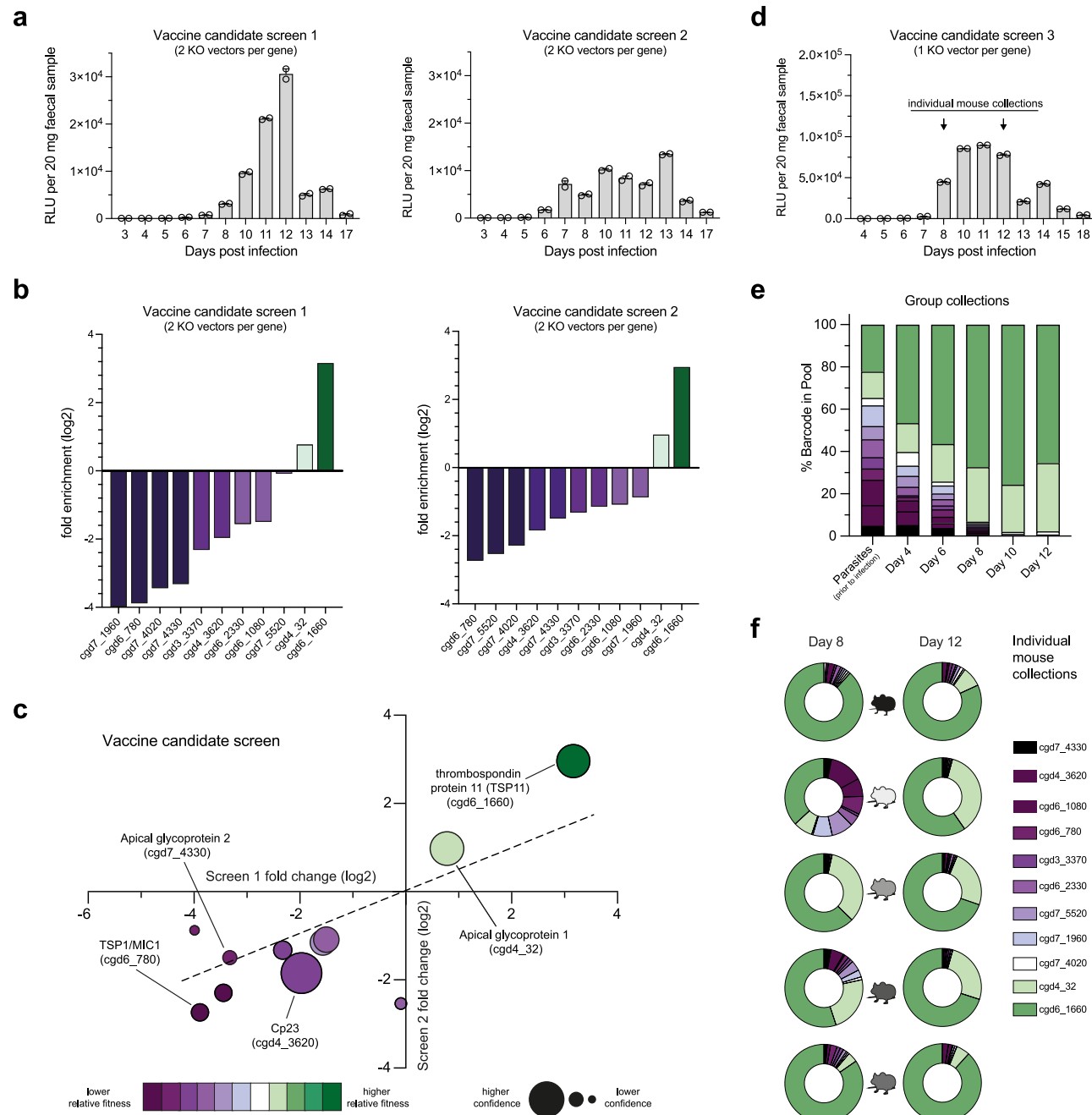

**Fig. 3 | In vivo CRISPR screen of *Cryptosporidium* vaccine candidates. a** Mouse faecal material was collected and luminescence was monitored during infection. Each replicate was conducted with 2 KO vectors per gene. Data shows the mean faecal luminescence from a pooled cage sample (± SEM of 2 technical replicates), $n = 5$ Ifnγ$^{-/-}$ mice for screen 1 and n = 3 Ifnγ$^{-/-}$ mice for screen 2. **b** Rank ordered fold enrichment scores from the vaccine candidate CRISPR screens. The colour indicates the relative fitness contribution of a gene, with dark purple being high fitness conferring and dark green being low fitness conferring. **c** Comparison of the fold enrichment scores between the replicate CRISPR screens for each gene. Confidence refers to the inverse of the 95% confidence interval when comparing the log2 fold change scores from each screen (see "Methods"). **d** Mouse faecal material was collected and luminescence was monitored during infection. Data shows the mean faecal luminescence from a pooled cage sample (± SEM of 2 technical replicates), $n = 5$ Ifnγ$^{-/-}$ mice. Barcodes from KO parasites could be easily monitored over time (**e**) and within individual mice (**f**).

close proximity to each other (Supplementary Fig. 7). Despite these mitigating factors, there is the potential for sublethal gene knockouts to exert synergistic or antagonistic effects and this possibility must be considered during the design process and while analysing and interpreting results.

Another important factor to consider is the amplifying effect of relative fitness screens within an in vivo model of infection. Genes that have a mild effect on parasite survival and fitness can be outcompeted within a larger pool. This appears to be the case for uracil phosphoribosyltransferase (UPRT, cgd1_1900), which catalyses the conversion of uracil and phosphoribosylpyrophosphate to uridine monophosphate. UPRT has recently been shown to be non-essential to parasite survival, yet the authors noted that their UPRT KO *Cryptosporidium* parasites appeared to have a growth defect[38]. In our in vivo screens, UPRT had a negative fold enrichment, indicating that UPRT contributes highly to parasite fitness and this likely reflects the observed growth defect. In contrast to UPRT, dCMP deaminase (cgd2_2780) had little effect on parasite fitness. dCMP deaminase

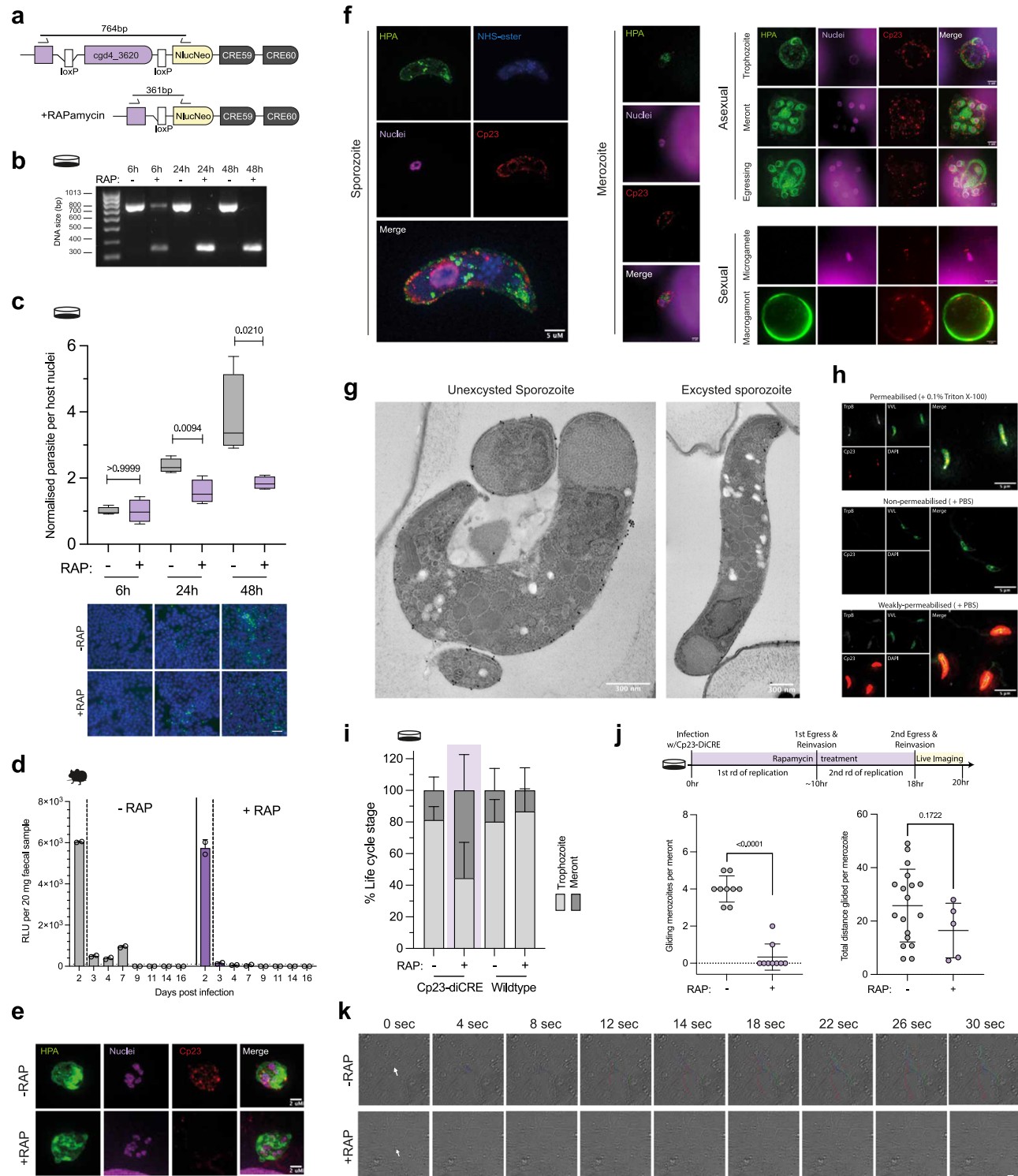

catalyses the conversion of deoxycytidine-monophosphate (dCMP) to deoxyuridine-monophosphate (dUMP), which is then transformed into deoxythymidine-monophosphate (dTMP) by DHFR-TS. Thymidine kinase can also produce dTMP, likely explaining why the loss of dCMP deaminase had little effect on parasite fitness.

Most of the genes in the vaccine candidate CRISPR screen were identified as fitness conferring, with the exception of apical glycoprotein 1 (AGP1) and thrombospondin repeat protein 11 (TSP11). Akey et al. previously showed AGP1 was dispensable for parasite survival and AGP2 was likely essential, both phenotypes that were recapitulated within our screens[32]. Recently, it has been suggested that there may be

redundancy in the *Cryptosporidium* thrombospondin protein family (TSPs), as there are 12 members, many with similar domain structures[39]. Although the *Cryptosporidium* TSPs currently have an unknown function, orthologous Apicomplexan TSP genes play a role in adhesion and motility. We included two TSPs in the vaccine candidate screen: (1) TSP8 (cgd6_780, also known as MIC1) and (2) TSP11 (cgd6_1660). TSP8/MIC1 is a known micronemal protein[40] and was identified as high fitness conferring. In contrast, TSP11 was dispensable, demonstrating that there is at least some redundancy within the *Cryptosporidium* TSP protein family. We also confirmed GP40 (cgd6_1080, also known as GP60) and GP900 (cgd7_4020), antigens

**Fig. 4 | Immunodominant antigen 23 is essential and required for reinvasion of host cells. a, b** HCT8 cell monolayers infected with Cp23-diCRE in the presence or absence of rapamycin. At 6, 24, and 48 h, gDNA was extracted and diagnostic PCRs confirmed the level of excision at the given timepoint. Data shown is representative of 2 biological replicates. **c** HCT8 cell monolayers infected with Cp23-diCRE in the presence or absence of rapamycin. At 6, 24, and 48 h, the monolayer was fixed and stained and the parasite per host nuclei was quantified. Box plot whiskers show the minimum and maximum values, the box shows the 25th to 75th percentile and the line shows the median. Representative images are shown, with nuclei in blue (Hoechst), and parasites in green (Vicia villosa lectin). Scale bar = 30 μm. Data shown is representative of two biological replicates. Significance was determined using a two-tailed unpaired *t* test. **d.** Ifnγ$^{-/-}$ mice were infected with 50,000 Cp23-diCRE parasites and treated with rapamycin or DMSO control in their drinking water at 2 days post infection. Data shows the mean faecal luminescence from a pooled cage sample ( ± SEM of 2 technical replicates), *n* = 2 mice per condition. **e** Super resolution microscopy at 24 h post infection in vitro with Cp23-diCRE parasites in the presence or absence of rapamycin; green (helix pomatia agglutinin (HPA)), parasite; magenta, (sytox), nuclei; red (Cp23 Ab), Cp23. Scale bar = 2 μm. Data shown is representative of two biological replicates. **f** Expansion microscopy of the *C. parvum* asexual stages: sporozoite, merozoite, trophozoite and meront, and the sexual stages: macrogamont (female) and microgamete (male); green (helix pomatia agglutinin (HPA)); blue (N-hydroxysuccinimide (NHS ester); magenta

(sytox); red (αCp23). Scale bar = 5 μm. Median expansion factor of 4.5. **g** Transmission electron microscopy of an excysted and unexcysted *C. parvum* sporozoite coupled with immunogold labelling with Cp23 antibody. Scale bar = 300 nm. **h** Permeabilisation assay of *C. parvum* sporozoites. The permeabilised condition used 0.1% Triton X-100 and the non-permeabilised condition used PBS. Grey (tryptophan synthase (αTrpB); green (Vicia villosa lectin); red (αCp23); blue (Hoechst). All images were taken using the same settings and exposure. Scale bar = 5 μm. Data shown is representative of two biological replicates. **i** HCT8 cell monolayers infected with either Cp23-diCRE or wildtype parasites in the presence or absence of rapamycin, and at 22 h post infection life stages were quantified. Data shows 2 biological replicates ± sd. *n* = 438 Cp23-diCRE - RAP, n = 173 Cp23-diCRE + RAP, *n* = 213 wildtype−RAP, *n* = 203 wildtype + RAP. **j, k** Live imaging of the reinvasion event was carried out at 18 h post infection of an HCT8 monolayer with Cp23-diCRE parasites in the presence or absence of rapamycin. All live microscopy data shown is from 2 biological replicates with *n* = 9 egress events recorded for each condition. Gliding distance was only measured for merozoites with clear movement following egress (*n* = 18 for DMSO and *n* = 5 for RAP). Movement of individual merozoites was tracked using Fiji software. Scatter plots show the mean ± sd with significance determined using a two-tailed unpaired t-test. Representative images are shown in (**k**) with movies in supplementary data. Scale bar = 8 μm. Note that a cell culture dish in the figure indicates an in vitro experiment, while a mouse silhouette indicates an in vivo experiment.

---

that are associated with protection from reinfection in humans, influence parasite fitness in vivo[25].

Cp23 is one of the leading cryptosporidiosis vaccine candidates, and here we demonstrated that it is highly fitness conferring, both in vitro and in vivo. Cp23 was first identified in 1986[33], and recent studies have revealed Cp23 recognising IgA and IgG are correlated with protection against *Cryptosporidium* infection in humans[25]. Until now, the function and precise localisation of Cp23 has been unclear. Here we validate a commercial Cp23 monoclonal antibody using diCRE mediated excision of the target gene, then use a combination of ultrastructure expansion, super-resolution, and transmission electron microscopy, to describe the localisation of Cp23 throughout the parasite life cycle at high resolution. Although our localisation agrees with previously published studies that detects Cp23 at the pellicle[41], our data suggests that Cp23 may not be exposed at the surface of the sporozoite. Cp23 has no detectable signal peptide, transmembrane domain or GPI-anchor, yet was recently demonstrated to be both myristoylated and palmitoylated[42]. Myristylation is a non-reversible lipid modifications that is commonly found in proteins that are anchored to internal membranes in other Apicomplexan parasites[43,44]. There are, however, some exceptions to this rule. TgMIC7 is a micronemal protein in *Toxoplasma* that is myristoylated prior to trafficking to the parasite surface[44]. Yet, TgMIC7 contains a transmembrane domain that likely facilitates entry into the secretory pathway, whereas Cp23 does not. While we cannot rule out trafficking of Cp23 to the outer membrane of the parasite, our data supports an internal localisation.

Despite this internal localisation, it is clear Cp23 IgA and IgG have both been correlated with protection in the clinic (Gilchrist et al.,[25]). One interpretation of this correlation is that antibodies recognising Cp23 arise from severe *Cryptosporidium* infection(s) where cell-based adaptive immunity against the parasite has developed. Controlled experiments using a natural murine model of *Cryptosporidium* suggest that antibody-based immunity is dispensable for resolution of infection[11]. Further, we found that the genetically validated Cp23 antibody used in this study was unable to block parasite attachment or invasion in vitro (Sup. Fig. 8). However, it has been reported that antibodies to Cp23 may have a protective effect in vivo, thus we cannot definitively rule out the possibility of Cp23-directed protection, whether through antibody or cell-based immunity[41]. Certainly, more investigation is warranted given we now know Cp23's essential role in parasite motility and survival.

With the targeted in vivo CRISPR screening method developed here, we assessed the fitness contributions of 22 *Cryptosporidium* genes. This number is near to the total number of *Cryptosporidium* genes with a genetically verified impact on parasite virulence or survival, prior to this study. As this screening technology develops and improves, we anticipate greater throughput, allowing for the assessment of a variety of phenotypes on a larger scale. This rapid assessment of phenotypes will allow us to expand the knowledge base for *Cryptosporidium* and explore basic biology that is essential for developing more effective interventions.

## Methods

All research in this manuscript was done in accordance with all relevant ethical regulations and animal work was approved by Home Office under project license PP8575470.

### Plasmid design and construction

Genomic sites for CRISPR directed repair were predicted using a customised version of EuPaGDT that retrieved 50 bp of flanking DNA surrounding the predicted PAM site[45]. Golden Gate Assembly or Gibson Assembly was used to generate all vectors used in this work. Golden Gate Assembly used, BsaI, BbsI-HF or BsmBI (New England BioLabs (NEB)) restriction enzymes. Gibson Assembly used HiFi DNA Assembly (NEB). To generate KO vectors for CRISPR screening, 2 consecutive Golden Gate reactions were performed. Firstly, between a Cas9-U6 vector and a 300 bp unique segment containing 50 bp homology arms (one of which contained the 50 bp gRNA), tracrRNA and a DNA barcode, creating a Cas9-unique plasmid. Secondly, between this Cas9-unique plasmid and an interchangeable selection cassette, in turn creating the KO vector. To generate diCRE mediated inducible knockouts, a gene of interest (GOI) was recodonised and cloned into a LoxP-Nluc-NeoR-diCRE vector via Gibson Assembly. To generate Cas9-U6-gRNA vectors, a 20 bp gRNA was cloned into the Cas9-U6-BsaI vector via Golden Gate Assembly[7,46].

### Culturing host cells

Human ileocecal adenocarcinoma cells (HCT8) were cultured in RPMI-1640 medium (Gibco) supplemented with 10% heat-inactivated foetal bovine serum (Merck), 120 U/mL penicillin (Life Technologies) and 0.1% amphotericin B (Gibco) at 37 °C under 5% $CO_2$. Cells were passaged at approximately 70% confluence using 0.25% trypsin-EDTA (Gibco). Cells were used for experiments between passage numbers 5

and 25. For super resolution and ultrastructure expansion microscopy, cells were seeded in 24-well plates containing coverslips. For high throughput microscopy quantifications, cells were seeded in black 96-well clear bottom tissue culture-treated plates (Corning). For transient transfections using luminescence, cells were seeded in 24-well tissue culture-treated plates (Corning). For live imaging, cells were seeded in μ-slide 8-well high chambers (Ibidi).

## Oocysts and excystation

*C. parvum* IOWAII strain oocysts were purchased from Bunch Grass Farm (Deary, ID). Oocysts were stored at 4 °C and used within 3 months of the date of isolation. The oocysts were excysted by incubating on ice with 1% sodium hypochlorite (VWR) in $H_2O$ for 5 min, followed by incubating with 0.75% sodium taurocholate (Merck) in RPMI-1640 medium with 1% foetal bovine serum at 37 °C (10 min for cell monolayer infections without transfection and 50 min prior for transfection). For in vitro infections, primed oocysts were used to infect HCT8 monolayers. To proceed with transfections, more than 60% of oocysts had to have excysted, observed using light microscopy at 50 min post sodium taurocholate treatment (Eclipse TS2R Nikon). For Supplementary Fig. 1, 0.75% sodium taurocholate was substituted with either 0.75% sodium deoxycholate (Sigma) or 0.75% sodium taurodeoxycholate (Sigma), and 1% RPMI-1640 was substituted with PBS.

## Generation of transgenic parasites

Excysted sporozoites were suspended in Lonza SF buffer, combined with the appropriate DNA and electroporated using programme 'FL115' on an AMAXA Nucleofactor 4D electroporator (Lonza) (FL115 was used for all transfections unless otherwise stated). For transient transfections in vitro, $5.0 \times 10^6$ oocysts were excysted and sporozoites were transfected in the 20 μL 16-well Nucleocuvette Strip format with 20 μg of plasmid. For generating stable transgenic parasites in vivo, $2.5 \times 10^7$ oocysts were excysted and sporozoites were transfected in the 100 μL Nucleocuvette Vessel format with 20 μg of Cas9-U6-gRNA plasmid and 40 μg of repair cassette containing the 50 bp homology arms (Supplementary Table 3 for all gRNA/homology primers used for individual genetic modifications in this study). For CRISPR screens, $5.0 \times 10^6$ sporozoites were excysted and sporozoites were transfected with 30 μg of KO vector for each gene (when 2 KO vectors were used, 15 μg of each KO vector was used). Transfections to knockout individual genes were performed separately (when 2 KO vectors per gene were used, transfections of KO vectors for the same gene were performed together). Parasites were pooled post transfection in 1% RPMI-1640 (Gibco) to infect mice and 4 or 5 mice were infected for each screen (Supplementary Tables 4–6 for all gRNA/homology primers used for individual genetic modifications in this study).

## Mouse model of infection

Interferon gamma deficient ($Ifn\gamma^{-/-}$) mice were bred and housed in pathogen-free conditions in the Biological Research Facility at The Francis Crick Institute. Mice of both sexes were used for experiments and mice were infected between the ages of 4wks to 8wks. To increase infection efficiency, mice were pretreated with an antibiotic cocktail: 1 g/L ampicillin (Merck), 1 g/L streptomycin (Merck) and 0.5 g/L vancomycin (Cambridge Biosciences) in their drinking water for 3–10 days prior to infection with transfected sporozoites. Before infection, mice received saturated sodium bicarbonate (Thermo Fisher Scientific) via oral gavage to neutralise their stomach acid. A second oral gavage was then undertaken 5 min thereafter with transfected sporozoites. Neither oral gavage exceeded the maximum volume of 0.1 mL per 10 g of body weight. For selection of transgenic parasites, 16 mg/mL paromomycin (BioServ) was administered in the mice's drinking water. To induce excision when diCRE-expressing transgenic parasites were used in vivo, 0.05 mg/mL rapamycin (Stratech Scientific) was administered in the mice's drinking water. No drug treatment lasted more than 3 consecutive weeks. During murine infections, faecal material was collected daily and stored at 4 °C. All experiments involving mice was done under the care and supervision of the Francis Crick Institute veterinary and Biological Research Facility staff, under protocols approved under project license PP8575470.

## Measuring parasite shedding by nanoluciferase

Faecal material was collected and 20 mg was lysed in faecal lysis buffer (50 mM Tris-hydrochloric acid, 10% glycerol, 1% Triton X-100, 2 mM dithiothreitol, 2 mM ethylenediaminetetraacetic acid (EDTA)). The lysate was clarified and combined with an equal volume of a 1:50 Nano-Glo Luciferase Assay Substrate: Nano-Glo Luciferase Assay Buffer (Promega). Luminescence was read at 200 gain on the BioTek Cytation5 (Agilent Technologies).

## Isolation of oocysts from mouse faeces

Faecal collections from the peak of infection were pooled, combined with cold water and filtered through a 250 mm mesh. The faecal suspension was then mixed 1:1 with saturated sucrose and pelleted by centrifugation at $1000 \times g$ for 10 min. Oocysts, located in the supernatant, were washed with cold water and pelleted by centrifugation at 1000 g for 5 min. A caesium gradient was used to isolate 750 μL of oocysts. Pure oocysts were washed with saline and stored for up to 6 months at 4 °C in saline.

## CRISPR screening barcoding

To obtain barcodes from the output, barcodes could either be extracted from the peak of infection or directly from daily faecal samples. To do so from the peak of infection, oocysts were purified from pooled faecal material (5–7 days surrounding the peak of infection), excysted and gDNA was extracted using the DNeasy Blood & Tissue Kit (Qiagen). To do so from daily faecal samples, DNA was extracted from 100 mg of faeces using the QIAamp PowerFecal Pro DNA Kit (Qiagen). Barcodes were obtained from the input material (100 μL of the pool of transfected parasites used to infect the mice) using the QIAquick PCR Purification Kit (Qiagen). Barcodes from the input and output were amplified with high fidelity KAPA polymerase (Roche) (25 cycles, annealing 68 °C and extension 15 seconds), and amplicon sequencing flanks were added with high fidelity KAPA polymerase (Roche) (10 cycles, 68 °C annealing and 15 seconds extension). The success of the nested PCR was confirmed via gel electrophoresis and barcodes were submitted for amplicon sequencing to The Genomics Science Technology Platform at The Francis Crick Institute. Illumina MiSeq platform with a paired-end 250 bp run configuration on a Nano flow cell was used.

## CRISPR screening analysis

The 7 bp barcode between the barcode primer binding sites was bioinformatically extracted from both input and output samples. Barcodes for each gene were counted and the percentage of the barcode in the pool calculated for both the input and output. Using the % barcode in the pool for the input and output, fold changes of all genes were calculated:

$$log_2 \, fold \, change = log_2 \left[ \frac{\% \, barcode \, in \, output}{\% \, barcode \, in \, input} \right]$$

## Immunofluorescence microscopy

At the required timepoint, HCT8 cell monolayers were washed with 1X PBS, fixed with 4% PFA/PBS (Alfa Aesar) for 15 min, permeabilised with 0.25% Triton X-100 (Merck) for 10 min and blocked with 4% BSA/PBS (Merck) overnight at 4 °C. Primary antibodies were incubated in 1% BSA/PBS for 2 h and then cell monolayers washed 5 times with 1X PBS. Secondary antibodies were incubated for 1 h in 1% BSA/PBS along with

the fluorescein labelled 1:4000 Vicia villosa lectin (VVL) (Vector Lab) or 1:1000 Helix pomatia agglutinin (HPA) (Invitrogen) that stains parasites. Nuclei were stained by incubating with 1:10,000 Hoechst 33342 (Invitrogen) in 1X PBS for 5 min. For the EdU assays, 10 mM EdU was incubated from 28 to 32 h post infection and stained as the described in the Click-iT EdU Cell Proliferation Kit for Imaging (C10340, Invitrogen). Stained monolayers were washed with 1X PBS, the coverslips mounted with ProLong Gold Antifade (Thermo Fisher Scientific) and visualised using a VisiTech instant super resolution imaging system (VT-iSIM). Alternatively, when super resolution was not required, the BioTek Cytation5 (Agilent Technologies) was used to visualise the stained monolayer. See Supplementary Table 7 for a full list of antibodies used in this work.

## Ultrastructure expansion microscopy

This protocol has been adapted from Liffner and Absalon, 2021. For the expansion of sporozoites, coverslips were coated with 0.1 mg/mL poly-D-lysine for 1 h and then washed twice with 1X PBS. Excysted sporozoites in 1% RPMI-1640 were added and allowed to adhere for 10 min at 37 °C. Adhered sporozoites or cell monolayers to expand were fixed with 4% PFA/PBS for 15 min at 37 °C. Protein crosslinking prevention was performed by adding 1.4% formaldehyde/ 2% acrylamide in 1X PBS to each sample, then incubating overnight at 37 °C. To perform gelation, TEMED and APS were added to a monomer solution (19% w/w sodium acrylate / 10% v/v acrylamide / 0.1% v/v N,N'-methylenebisacrylamide in 1X PBS) and pipetted under each coverslip in a pre-cooled humid chamber which was then incubated for 5 min on ice. To complete gelation, the humid chamber was incubated for 1 h at 37 °C. To separate gels from the coverslips, the coverslips were incubated with a denaturation buffer (200 mM SDS, 200 mM NaCl, 50 mM Tris in water, pH 9) for 15 min. To complete the denaturation of the sample, the gel was incubated in a denaturation buffer for 90 min at 95 °C. To expand the sample, the gel was incubated with $dH_2O$ for 30 min, three times in total at room temperature (RT). Following the first round of expansion, the sample was shrunk by incubating with 1X PBS, 2 times in total at RT. To block the sample, 2% BSA/PBS was incubated for 1 h at RT. To stain the sample, primary antibodies were incubated in 2% BSA/PBS overnight at RT. Gels were then washed 3 times in total with 0.5% Tween 20/PBS for 10 min. Directly conjugated and secondary antibodies were incubated in 1X PBS for 3 h at RT. The gel was then washed 3 times in total with 0.5% Tween20/PBS for 10 min. A second round of expansion took place by incubating the gel with $dH_2O$ for 30 min, three times at RT. The gel was measured to calculate the expansion factor, mounted onto a 0.1 mg/mL poly-D-lysine coated 60 mm dish and visualised on the VT-iSIM microscope.

## Immunogold electron microscopy

$1 \times 10^7$ oocysts were excysted for 1 h, pelleted and re-suspended in fixative (8% formaldehyde in 0.4 M HEPES buffer, pH 7.4) for 15 min at RT. Samples were washed with 0.2 M HEPES and a secondary fixation (2% formaldehyde + 0.05% glutaraldehyde in 0.2 M HEPES) step was carried out at 4 °C overnight. Samples were dehydrated and infiltrated with LR white resin at −20 °C overnight. Prior to polymerisation, samples were brought back to RT for 1 h and then polymerised at 60 °C for 24 h. The samples were sectioned using a Leica UC7 ultramicrotome with a 45° Diatome diamond knife, achieving sections of 70 nm thickness. The sections were collected on nickel grids and immunogold labelled. To do so, the samples were quenched with PBS/glycine for 2 min three times and then blocked with 1% BSA/PBS for 5 min at RT. A 1:10 dilution of the primary Cp23 antibody was incubated in 1% BSA/PBS for 1 h and then samples were washed twice with 0.1% BSA/PBS. A 1:50 dilution of the protein-A gold bound to 10 nm gold particles (PAG-10) was incubated with the sample in 0.1% BSA/PBS for 20 min and then washed twice with PBS. A post fixation step was carried out with 1% glutaraldehyde for 5 min and samples were washed

with Milli-Q $H_2O$ for 1 min 6 times in total. Samples were incubated in 1% uranyl acetate for 10 min and air dried. Transmission electron microscopy was carried out on the 120-kV JEOL JEM-1400Flash Electron Microscope (JEOL Ltd., Welwyn Garden City, UK) with a JEOL Matataki Flash camera.

## Live microscopy

At 18 h post-infection, phase contrast imaging was performed using an Eclipse Ts2R microscope (Nikon) with a 40X/0.55 NA Ph1 ADL objective (Nikon), digital sight 10 camera (Nikon) and a TPi-TCSX (Tokai Hit) heated stage set at 37 °C. Images were acquired at 25fps for 1 h to capture egressing merozoites. Images were imported into ImageJ2 Version 2 and the number of gliding merozoites was manually observed. Image sequences were converted to movies to acquire short videos of egressing merozoites.

## Permeabilisation assay

Coverslips placed in 24-well plates were coated with 0.1 mg/mL poly-D-lysine for 1 h and washed twice with 1X PBS. Excysted sporozoites in serum free RPMI-1640 were allowed to adhere for 10 min at 37 °C. Adhered sporozoites were fixed with 1% PFA/PBS for 20 min at RT. For the permeabilised condition, 0.1% Triton X-100 was used to permeabilise for 10 min at RT. For the non-permeabilised condition, 1X PBS was instead incubated for 10 min at RT. Both conditions were blocked with 4% BSA/PBS overnight at 4 °C. Primary antibodies were incubated in 1% BSA/PBS for 2 h at RT. The coverslips were washed 5 times and secondary antibodies were incubated in 1% BSA/PBS for 1 h at RT. All coverslips were mounted with ProLong Gold Antifade (Thermo Fisher Scientific) and visualised using a VT-iSIM.

## Attachment assay

Coverslips placed in 24-well plates were coated with 0.025 mg/mL poly-D-lysine for 1 h and washed twice with 1X PBS. 60,000 oocysts per well were excysted and sporozoites were added and spun at 80 g for 1 min to adhere. Attachment was performed in Ringer's solution (10 mM Hepes pH 6.7, 10 mM Glucose, 2 mM $CaCl_2$, 1 mM $MgCl_2$, 3 mM KCl, 3 mM $NaH_2PO_4$ and 155 mM NaCl) for 15 min at 37 °C. Sporozoites were fixed using 8% PFA/PBS leak in resulting in 4% PFA/PBS/Ringer's fixation for 15 min at RT. Sporozoites were washed 3 times in total with 1X PBS and blocked with 4% BSA/PBS overnight at 4 °C. Sporozoites were stained with 1:5000 Helix pomatia agglutinin (HPA) in 1% BSA/PBS for 1 h at RT. The BioTek Cytation5 (Agilent Technologies) was used to visualise the attached sporozoites. Images were exported as TIFs into ImageJ2 Version 2 and the below macro was used to analyse the data:
run("Subtract Background...", "rolling=50");
setAutoThreshold("Default dark");
//run("Threshold...");
setAutoThreshold("Triangle dark");
setOption("BlackBackground", true);
run("Convert to Mask");
run("Median...", "radius=2"); run("Analyze Particles...", "size=50-10000 circularity=0.00-1 display summarize overlay");

## Neutralisation assay

HCT8 cells were seeded to confluency in 96-well clear bottom tissue culture-treated plates (Corning). An 8-point twofold dilution series of the Cp23 antibody (Stratech, LS-C137378) and the IgG isotype control Stratech, GTX35009 + 0.02% Proclin 300 (Merck, 48912-U) was prepared (0.02 mg/mL–0.0002 mg/mL) in 96-well non-tissue culture-treated plates. 25,000 oocysts per well were excysted and incubated with the antibody dilution series for 5 min at 37 °C. Post antibody incubation, the sporozoites were added to the 96-well plate containing HCT8 cells in 1% RPMI-1640. At 24 h post-infection, cell monolayers were washed with 1X PBS, fixed with 4% PFA/PBS (Alfa Aesar) for 15 min, permeabilised with 0.25% Triton X-100 (Sigma) for 10 min and blocked

with 4% BSA/PBS (Merck) overnight at 4 °C. Parasites were stained with 1:4000 Vicia villosa lectin (VVL) and host nuclei were stained with 1:10,000 Hoechst 33342 (Invitrogen) in 1% BSA/PBS for 1 h at RT. Cell monolayers were washed and visualised on the BioTek Cytation5 (Agilent Technologies). 2 × 2 tiled images per well were acquired with the ×20 objective. The Gen5 analysis software (Agilent Technologies) was used to count the number of host cell nuclei and parasites for parasite per nuclei calculations.

### Reporting summary
Further information on research design is available in the Nature Portfolio Reporting Summary linked to this article.

## Data availability
All raw data used for graphs is available in the Source Data file. Source data are provided with this paper.

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

## Acknowledgements

We thank Pippa Hawes of The Francis Crick Electron Microscopy Scientific Technology Platform for her helpful discussions and Elena Rodrigues of the Cryptosporidiosis Laboratory for preparing *Cryptosporidium* samples for electron microscopy. We also thank Nicholas Chisholm and other members of the Biological Research Facility, as well as the Genomics Scientific Technology Platform at The Francis Crick Institute for their contributions to this work, and Rodrigo Baptista of Houston Methodist for his insight and helpful conversations. Lastly, we thank VEuPathDB for maintenance of critical databases that were essential for this work. This work was supported by the Francis Crick Institute, which receives its core funding from Cancer Research UK, the Medical Research Council, and the Wellcome Trust (CC2063), and by a UKRI grant awarded to A.S. (101042783).

## Funding

## Competing interests

The authors declare no competing interests.
