## [Transparent Peer Review file · Nature Communications]

Targeted CRISPR Screens Reveal Genes Essential for *Cryptosporidium* Survival in the Host Intestine

Corresponding Author: Mr Adam Sateriale

Version 0:

Reviewer comments:

Reviewer #1

(Remarks to the Author)

Lucy C. Watson et al. developed a targeted CRISPR-based screening method to investigate parasite genes that affect parasite survival in vivo. They used this method to screen the parasite's pyrimidine rescue pathway and several vaccine candidates of *Cryptosporidium*. Through screening, they found that Cp23 has high fitness conferring, suggesting that it may be important for parasite survival. Furthermore, they found that Cp23 is essential for parasite gliding and reinfection in vitro by optimized diCRE-mediated genetic editing.

Overall, the manuscript seems to contribute to the exploration of a rapid way to screen the essential genes in *C. parvum* for parasite growth. The authors may find the following specific points useful in improving the manuscript.

1. It would be better to show each individual data in the panel, such as Figures 1b, 1e, et al. Currently, it is hard to know how many replicates you used in each experiment.
2. There were some partial excisions in the RNR-diCRE parasites at 24 and 48 hours after adding RAP (Figure 2b). Why did you get different excision results from PCR analysis when using the diCRE system for different genes in *C. parvum*? Is it possible that the DiCre system is not suitable for all essential genes in *C. parvum*? Could you clarify the reason for these results?
3. Have you tried knocking out Cp23 directly in *C. parvum*? If Cp23 is essential, it should not be possible to knock it out directly in *C. parvum*. Now you showed that Cp23 is essential for parasite reinvasion of the host cell. If it is essential, it should be important for the initial invasion of sporozoites released from oocysts.
4. Is there a commercial Cp23 antibody? I didn't see any details about this antibody in the method. The U-ExM showed that Cp23 not only localizes to the sporozoite surface but also around the parasite nuclei. To get a better answer about the localization of Cp23, the construction of a Cp23 endogenous tagging strain might be a good choice.
5. In Figure 4e, you used the Cp23 antibody to stain the parasite. I was wondering at this time what the ratio of Cp23 positive parasites compared to the no-treatment group was after adding RAP for 20 hours. The result would support the data in Figure 4k. Now, it is difficult to know in the RAP treatment group if this parasite shown in the live image is the CP23 knockout parasite.

Reviewer #2

(Remarks to the Author)

This is an outstanding article that is a significant advance for the cryptosporidium research community, both in terms of new techniques development but also in terms of the new data of high biological and translational importance. The authors demonstrate a method to interrogate at least 11 genes at a time for their functional significance by simultaneous CRISPR/Cas9 knockouts with barcoded genes and subsequent sequencing to get relative fitness levels. This can be done in vitro or in vivo (mouse passage). They go on to use this to assess 11 genes in pyrimidine metabolism and also 11 genes that represent potential vaccine candidates. The findings in both cases, are of great biological and translational significance.

They also refine the conditional expression system, diCRE, such that the leakiness of the system is fixed. This is used to further study the "essential genes" and their functions, found in the genetic screens. I have only minor editorial and display comments, as the article is very clear and well-written.

1. Page 3, lines 59-63. "Nitazoxanide... is not approved for use in children" In fact, Nitazoxanide is approved for use in children down to 1 years of age. Since the major morbidity and mortality is in children from 6 to 18 mos of age, some very young children are not covered by the indication. But more importantly, Amadi and colleagues have shown that Nitazoxanide is only about 30% efficacious (subtracting the "response" from placebo) in malnourished children. Thus a much more effective drug is needed for this population, and malnourished children represent the majority of the need throughout the world. Perhaps the authors could refine this statement to be correct?
2. Page 4, line 85 streamlined instead of "streamline"
3. Page 5 line 141: Since this is the data for a section in the paper, it would be better to move one or two parts into regular display figures. I suggest Sup 1d and 1e.
4. Fig 1, Fig 2, Fig 3, & Fig 4 are very difficult to read, and Fig 2 is a bit difficult too. I had to enlarge to 150% - 200% and use magnifier to read. I highly suggest enlarging, thus breaking some into two figures and maybe moving some parts of the figs into the supplementary. I know you are limited to "10 figures and tables", but I would move Tables 3-6 into supplementary data to make room.
5. Page 8, line 221: Please explain the significance of the T2A skip peptide, so the general reader can understand why you reasoned it may not be functioning properly.
6. Page 7-8 203-234 and Supplementary Figures 2&3: Again, it would be good to have one display figure for this section in the regular MS, as it's a major point of the paper (could be a summary figure of the amount of improvement overall?).
7. Page 12, line 360: You mention 30bp works in your hands. It bears repeating here that larger flanks lead to more efficient targeting, as per results (at least to 50bp and possibly longer)
8. Page 31, Fig 1 legend: Point out in this Figure 1 where there's a mouse picture, it represents in vivo data derived by passaging the parasites through a mouse.
9. Page 33, Fig 2 legend: Point out that in this Figure 2 and others following that TC plates pictures on figures mean results reflect experiments carried out in vitro culture but where there's a mouse picture, it represents in vivo data. Probably need to state in first supplementary figure too.
- 10 Page 37, Suppl Fig 1d: What do the 3 small panels at the top and the large panel at the bottom mean for the left and right parts of Supp Fig 1d represent? It is difficult for this reviewer to see any difference between the left and right panels that we are told are different. Perhaps just omit Supp Fig 1d as the Luminometer measurements are sufficient and quantitative.

Reviewer #3

(Remarks to the Author)

The manuscript by Watson et al. describes refinements in the protocols to increase transfection efficiency, and modification of existing vectors for gene essentiality and inducible gene knockout experiments in *Cryptosporidium*. The authors conducted pilot CRISPR screens to test the essentiality of 11 genes in the pyrimidine salvage pathway, and 10 vaccine candidate genes. They validated pyrimidine salvage genes already known to be dispensable and identified others that were high/low fitness conferring for the parasite. Similarly, for the small set of vaccine candidates, they validated known dispensable genes and identified other genes that may be essential for parasite survival. Although these methodological refinements will be helpful for researchers performing genetic manipulation studies in *Cryptosporidium*, the authors have overstated the novelty of their CRISPR screen for assessing gene essentiality and the cre-lox approach for generating inducible gene knockouts. The authors have simply modified the available CRISPR vector to include the Cas9/guide and repair in one construct along with a barcode, performed individual transfections per targeted gene, and then pooled parasites post-transfection for their small-scale CRISPR gene essentiality screen. Moreover, the leakiness reported with the recently published cre-lox approach is in vitro and not in vivo, and minor modifications were made to the existing cre-lox approach. The authors show localization of CP23 (vaccine candidate) in sporozoite, asexual, and sexual stages and found that merozoites lacking CP23 were defective in gliding motility and re-invasion. However, they have not assessed the role of CP23 in sexual stages of the parasite's lifecycle since the protein is expressed throughout the parasite lifecycle based on its localization. Additional experiments and phenotypic analysis are required to understand the molecular function of CP23 during asexual and sexual stages.

Major comments:

Title is not reflective of the pilot/small scale CRISPR screening data presented.

Using the non-deoxy form of sodium taurocholate, along with other minor modifications increased the current transfection efficiency by more than 50-fold (Sup Fig. 1). Luminescence readings rather than normalized luminescence units, and biological replicates need to be provided to support this key finding of this manuscript.

It is already known that 50 bp homology arms is sufficient for repair, and therefore the data presented does not add any new information.

For the pilot CRISPR screening, additional repeats with multiple biological replicates are required to confirm the fitness conferring outcome.

Additional experiments are required to fully understand the function of CP23. What is its role in sexual stages since it is expressed in these stages as well as shown in Fig.4f?

Results of Triton-X experiments (Fig. 4h) are not clear, and are actually contrasting with the ultrastructure expansion microscopy findings that show localization on parasite pellicle with permeabilization.

Imaging panels provided in Fig. S4e for +/- rapamycin experiments are too small to see any differences in parasites with CP23 staining.

Details on age of mice used for infection experiments are lacking.

Abstract: It is not clear how this small scale screen is a 'highly versatile' method and will aid to 'more rapidly' expand *Cryptosporidium* biology

Minor comments: There are many typos in the manuscript:

Change RMPI to RPMI

Change optimise to optimize

Change synthesise to synthesize

Version 1:

Reviewer comments:

Reviewer #1

(Remarks to the Author)

The authors have satisfactorily addressed and clarified all concerns in the revised manuscript. I have no further comments on the study.

Reviewer #2

(Remarks to the Author)

The authors have addressed all of my concerns and it appears have well-addressed the concerns of other reviewers.

Reviewer #3

(Remarks to the Author)

The authors have fixed the minor concerns in the revised version, but still many of the major concerns raised earlier have not been addressed.

No additional experiments were conducted as suggested in the revised manuscript to understand the role of CP23 during the parasite lifecycle. Although, gamete fertilization is blocked in HCT-8 cells, sexual stages can be observed in HCT-8 cells and phenotypic defects upon CP23 deletion can be assessed. These experiments are necessary since the data provided on CP23 function is not conclusive.

One of the major points of the paper is the 50-fold increase in transfection efficiency. As the authors have indicated in their response, there is variability in transfection depending on the fresh parasite batch; therefore, it is pertinent to show the raw luminescence data for the readers to fully understand the variability between assays.

The authors have provided a repeat of their pilot CRISPR screen (Fig. S2), but the results are not consistent and appear somewhat different. For example, why does *cgd1_1900* show contrasting results in terms of fitness in the repeat experiment?

CP23 localization data is still unclear and requires quantification and showing additional images for readers to fully understand the changes in localization of this protein under different permeabilization conditions. The authors have still not addressed the question of why the CP23 staining in permeabilized parasites for the expanded parasites in Fig.4F appears different from the permeabilized parasite panel in Fig.4H.

Version 2:

Reviewer comments:

Reviewer #3

(Remarks to the Author)

The authors have addressed the concerns.

REVIEWER COMMENTS

Reviewer #1 (Remarks to the Author):

Lucy C. Watson et al. developed a targeted CRISPR-based screening method to investigate parasite genes that affect parasite survival in vivo. They used this method to screen the parasite's pyrimidine rescue pathway and several vaccine candidates of *Cryptosporidium*. Through screening, they found that Cp23 has high fitness conferring, suggesting that it may be important for parasite survival. Furthermore, they found that Cp23 is essential for parasite gliding and reinfection in vitro by optimized diCRE-mediated genetic editing.

Overall, the manuscript seems to contribute to the exploration of a rapid way to screen the essential genes in *C. parvum* for parasite growth. The authors may find the following specific points useful in improving the manuscript.

We thank the reviewer for their time and providing their specific points to improve the manuscript. We've done our best to implement them.

1. It would be better to show each individual data in the panel, such as Figures 1b, 1e, et al. Currently, it is hard to know how many replicates you used in each experiment.

Sadly, these initial experiments did not have individual mouse collections. These are from pooled faecal collections from the cage, which is a common method used in the cryptosporidium field (Viniyak 2015, Nature). We have adjusted the wording in all figure legends to make this clearer to the reader: 'Data shows the mean faecal luminescence from a pooled cage sample (\pm SEM of 2 technical replicates), n = x number of mice'

2. There were some partial excisions in the RNR-diCRE parasites at 24 and 48 hours after adding RAP (Figure 2b). Why did you get different excision results from PCR analysis when using the diCRE system for different genes in *C. parvum*? Is it possible that the DiCre system is not suitable for all essential genes in *C. parvum*? Could you clarify the reason for these results?

Well spotted by the reviewer. We believe this has more to do with our PCR analysis. Parasites that die will likely have their DNA degraded, whereas parasites that survive will not. So, if even 1% of parasites escape rapamycin induced excision, their DNA will be overrepresented during the PCR analysis because parasites with successful excision will have died.

3. Have you tried knocking out Cp23 directly in *C. parvum*? If Cp23 is essential, it should not be possible to knock it out directly in *C. parvum*. Now you showed that Cp23 is essential for parasite reinvasion of the host cell. If it is essential, it should be important for the initial invasion of sporozoites released from oocysts.

In response to their reviewer's suggestion, we have performed this experiment and included it as supplementary figure 5. In line with our previous results, we were not

able to KO Cp23, in contrast to the TK KO positive control transfected and selected at the same time. Unfortunately, with the current models and genetic tools, we are not able to test the effect of Cp23 in sporozoites. Rapamycin induced excision takes around 12-24 hours to exert an effect on the protein level (as we've shown). Therefore, to test the effect on sporozoites, we would need to remove the protein from oocysts by treating mice, but as we've shown this protein is essential for the parasite to survive in mice.

4. Is there a commercial Cp23 antibody? I didn't see any details about this antibody in the method. The U-ExM showed that Cp23 not only localizes to the sporozoite surface but also around the parasite nuclei. To get a better answer about the localization of Cp23, the construction of a Cp23 endogenous tagging strain might be a good choice.

What we have used is a commercial Cp23, which is target validated through our inducible knockout line. In the methods section we have included the catalogue and lot number for readers. We have indeed tried to create a C-terminal HA-labelled Cp23 parasite line and failed – data that we are including as a supplementary figure 5 and have noted in the manuscript: We believe this failure is likely because the C-terminal end is essential for Cp23 function and cannot withstand an epitope tag.

5. In Figure 4e, you used the Cp23 antibody to stain the parasite. I was wondering at this time what the ratio of Cp23 positive parasites compared to the no-treatment group was after adding RAP for 20 hours. The result would support the data in Figure 4k. Now, it is difficult to know in the RAP treatment group if this parasite shown in the live image is the CP23 knockout parasite.

Unfortunately we did not record ratios for the experiment in 4e, but we believe the data the reviewer is looking for is in 4i – with rapamycin treatment you see a higher ration of late stage meronts to newly invaded (trophozoites).. this suggests that the motile stages are unable to reinvade, and (as the reviewer notes) this supports the data in Figure 4k.

Reviewer #2 (Remarks to the Author):

This is an outstanding article that is a significant advance for the cryptosporidium research community, both in terms of new techniques development but also in terms of the new data of high biological and translational importance. The authors demonstrate a method to interrogate at least 11 genes at a time for their functional significance by simultaneous CRISPR/Cas9 knockouts with barcoded genes and subsequent sequencing to get relative fitness levels. This can be done in vitro or in vivo (mouse passage). They go on to use this to assess 11 genes in pyrimidine metabolism and also 11 genes that represent potential vaccine candidates. The findings in both cases, are of great biological and translational significance. They also refine the conditional expression system, diCRE, such that the leakiness of the system is fixed. This is used to further study the "essential genes" and their functions, found in the genetic screens. I have only minor editorial and display comments, as the article is very clear and well-written.

Thank you to the reviewer for their time and kind words.

1. Page 3, lines 59-63. "Nitazoxanide... is not approved for use in children" In fact, Nitazoxanide is approved for use in children down to 1 years of age. Since the major morbidity and mortality is in children from 6 to 18 mos of age, some very young children are not covered by the indication. But more importantly, Amadi and colleagues have shown that Nitazoxanide is only about 30% efficacious (subtracting the "response" from placebo) in malnourished children. Thus a much more effective drug is needed for this population, and malnourished children represent the majority of the need throughout the world. Perhaps the authors could refine this statement to be correct?

We have amended the wording to 'only partially effective in children'. Thank you to the reviewer for clarifying.

2. Page 4, line 85 streamlined instead of "streamline"

Amended in the text.

3. Page 5 line 141: Since this is the data for a section in the paper, it would be better to move one or two parts into regular display figures. I suggest Sup 1d and 1e.

We thank the reviewer for this suggestion. What we've included in the supplementary figure is the protocol optimisation experiments. We've done this, because 1) we want the focus to be on the biology 2) this layout was strongly encouraged from non-cryptosporidium experts who gave us feedback prior to submitting the manuscript. We appreciate that the reviewer is likely a Cryptosporidium expert and excited about our protocol improvements (we are too!), but (respectfully) we would like the focus of this manuscript to remain on the biology and not protocol optimisation.

4. Fig 1, Fig 2, Fig 3, & Fig 4 are very difficult to read, and Fig 2 is a bit difficult too. I had to enlarge to 150% - 200% and use magnifier to read. I highly suggest enlarging, thus breaking some into two figures and maybe moving some parts of the figs into the supplementary. I know you are limited to "10 figures and tables", but I would move Tables 3-6 into supplementary data to make room.

We have moved the tables into supplementary and will consider adjusting the formatting. Thank you for the suggestion.

5. Page 8, line 221: Please explain the significance of the T2A skip peptide, so the general reader can understand why you reasoned it may not be functioning properly.

This is now amended in the text.

6. Page 7-8 203-234 and Supplementary Figures 2&3: Again, it would be good to have one display figure for this section in the regular MS, as it's a major point of the paper (could be a summary figure of the amount of improvement overall?).

Please see suggestion above.

7. Page 12, line 360: You mention 30bp works in your hands. It bears repeating here that larger flanks lead to more efficient targeting, as per results (at least to 50bp and possibly longer)

We've noted this in the text.

8. Page 31, Fig 1 legend: Point out in this Figure 1 where there's a mouse picture, it represents in vivo data derived by passaging the parasites through a mouse.

We have removed these guides from all figures where we only use only one type of experiment and this is made clear from the figure legend. We have added the following statement to Figures 2 and 4, where we have mixed experiments: 'Note that a cell culture dish in the figure indicates an in vitro experiment, while a mouse silhouette indicates an in vivo experiment.' We hope this will clarify for the reader – thank you for pointing this out.

9. Page 33, Fig 2 legend: Point out that in this Figure 2 and others following that TC plates pictures on figures mean results reflect experiments carried out in vitro culture but where there's a mouse picture, it represents in vivo data. Probably need to state in first supplementary figure too.

See above.

10 Page 37, Suppl Fig 1d: What do the 3 small panels at the top and the large panel at the bottom mean for the left and right parts of Supp Fig 1d represent? It is difficult for this reviewer to see any difference between the left and right panels that we are told are different. Perhaps just omit Supp Fig 1d as the Luminometer measurements are sufficient and quantitative.

The reviewer is completely right, we did not sufficiently label this figure. The three small panels represent different fluorophores and give the reader a visual representation of the percentage of transfected parasites (something that is lost in luminescence readings alone) We have added these to the figure to clarify for the reader.

Reviewer #3 (Remarks to the Author):

The manuscript by Watson et al. describes refinements in the protocols to increase transfection efficiency, and modification of existing vectors for gene essentiality and inducible gene knockout experiments in *Cryptosporidium*. The authors conducted pilot CRISPR screens to test the essentiality of 11 genes in the pyrimidine salvage pathway, and 10 vaccine candidate genes. They validated pyrimidine salvage genes already known to be dispensable and identified others that were high/low fitness conferring for the parasite. Similarly, for the small set of vaccine candidates, they validated known dispensable genes and identified other genes that may be essential for parasite survival. Although these methodological refinements will be helpful for researchers performing genetic manipulation studies in *Cryptosporidium*, the authors have overstated the novelty of their CRISPR screen for assessing gene essentiality and the cre-lox approach for generating inducible gene knockouts. The authors have

simply modified the available CRISPR vector to include the Cas9/guide and repair in one construct along with a barcode, performed individual transfections per targeted gene, and then pooled parasites post-transfection for their small-scale CRISPR gene essentiality screen. Moreover, the leakiness reported with the recently published cre-lox approach is in vitro and not in vivo, and minor modifications were made to the existing cre-lox approach. The authors show localization of CP23 (vaccine candidate) in sporozoite, asexual, and sexual stages and found that merozoites lacking CP23 were defective in gliding motility and re-invasion. However, they have not assessed the role of CP23 in sexual stages of the parasite's lifecycle since the protein is expressed throughout the parasite lifecycle based on its localization. Additional experiments and phenotypic analysis are required to understand the molecular function of CP23 during asexual and sexual stages.

We thank the reviewer for their time and comments, which we have strived to implement into the revised manuscript.

Major comments:

Title is not reflective of the pilot/small scale CRISPR screening data presented.

We respectfully disagree with the reviewer here. We believe the term 'targeted' is apt to describe a method where you can target a certain pathway or gene family to identify essential genes. Further, we have deliberately refrained from using 'high-throughput' or other wording that would oversell the method and/or confuse the reader.

Using the non-deoxy form of sodium taurocholate, along with other minor modifications increased the current transfection efficiency by more than 50-fold (Sup Fig. 1). Luminescence readings rather than normalized luminescence units, and biological replicates need to be provided to support this key finding of this manuscript.

It is our fault this wasn't presented more clearly. These data are repeated, which is why normalised luminescence units were required. The transfectability of *Cryptosporidium* parasites changes over time – freshly isolated parasites are more amenable to transfection than older parasites and transfectability changes between bovine faecal isolation shipments. Again, there is no continuous cell culture for this parasite which is why this work is so technically challenging. Using normalised luminescence units is the only way of effectively combining experiments because of this. We've adjusted the wording to make this clearer to the reader and we thank the reviewer for noting this.

It is already known that 50 bp homology arms is sufficient for repair, and therefore the data presented does not add any new information.

We believe this does provide new information, as this study systematically tests the length of homology arms required for insertion in *Cryptosporidium*. We opted to keep this data as a supplementary figure in the updated manuscript.

For the pilot CRISPR screening, additional repeats with multiple biological replicates are required to confirm the fitness conferring outcome.

At the reviewer's suggestion, we have repeated the pilot CRISPR screen and included this data as supplementary figure 2. In the draft manuscript we compared the reproducibility of the pilot results using 1gRNA vs 2gRNA, which demonstrated that results were consistent with either 1 or 2 guides. We have now repeated the 1gRNA pilot screen here and, again, shown that the data is consistent.

Additional experiments are required to fully understand the function of CP23. What is its role in sexual stages since it is expressed in these stages as well as shown in Fig.4f?

Unfortunately, we are unable to fully study sexual stages within the current experimental systems for *Cryptosporidium*. Within cell culture, *Cryptosporidium* replicates asexually and then develops into sexual stages. However, these sexual stages do not produce new oocysts. This makes it rather difficult to study protein function as you are working with stage of the parasite that are clearly defective. While we agree with the reviewer that the function of Cp23 in sexual stages will certainly be interesting, we believe is it highly difficult with the current model systems and well beyond the scope of this particular manuscript.

Results of Triton-X experiments (Fig. 4h) are not clear, and are actually contrasting with the ultrastructure expansion microscopy findings that show localization on parasite pellicle with permeabilization.

We believe this is addressed within the manuscript – lines 317-325: 'When permeabilised with Triton X-100, sporozoites demonstrated faint staining with the Cp23 antibody (Fig. 4h). Without permeabilisation, *C. parvum* sporozoites, surprisingly, demonstrated either a complete lack of Cp23 signal or a heightened intensity of signal throughout the parasite (Fig. 4h). In these high intensity parasites, we can detect the internal control antibody (CpTrpB - *Cryptosporidium* tryptophan synthase beta) at low levels. This suggests that 1) high intensity Cp23 parasites are weakly permeabilised, 2) permeabilisation with Triton X-100 affects Cp23 localisation, likely by disrupting its membrane association, and 3) Cp23 is likely not exposed on the surface of the sporozoite.'

Imaging panels provided in Fig. S4e for +/- rapamycin experiments are too small to see any differences in parasites with CP23 staining.

We have adjusted these figures for clarity.

Details on age of mice used for infection experiments are lacking.

We have amended this in the methods section to include the age range of mice used in this study.

Abstract: It is not clear how this small scale screen is a 'highly versatile' method and will aid to 'more rapidly' expand *Cryptosporidium* biology

We believe the screen is highly versatile because the payload that is delivered in the genetic screen can be anything... a stage reporter, a fluorescent marker, or a luminescent reporter as we've used in this manuscript. This allows researchers to

construct any genetic screen they can imagine or even implement new reporters as they are developed. We also believe that being able to quantify the fitness of multiple genes is an improvement over the current technology, which is to assess the fitness of each gene on a case-by-case basis. We are aware that, compared to other systems, our screens are small-scale, but they are still a clear and significant improvement over the current technology as these are the first genetic screens in *Cryptosporidium*.

Minor comments: There are many typos in the manuscript:

Change RMPI to RPMI

Change optimise to optimize

Change synthesise to synthesize

Thank you to the reviewer for catching RMPI – this has been amended. Optimise and synthesis are the preferred spellings in the UK and Nature is a UK-based publication.

The authors have fixed the minor concerns in the revised version, but still many of the major concerns raised earlier have not been addressed.

No additional experiments were conducted as suggested in the revised manuscript to understand the role of CP23 during the parasite lifecycle. Although, gamete fertilization is blocked in HCT-8 cells, sexual stages can be observed in HCT-8 cells and phenotypic defects upon CP23 deletion can be assessed. These experiments are necessary since the data provided on CP23 function is not conclusive.

We respect the reviewer's request, but we still feel that 1) these experiments are not within the scope of this manuscript and 2) these experiments are not feasible or a wise use of resources.

In cryptosporidium infected cell culture, you do have male and female gametes, but fertilisation is blocked and the reason for this block is unknown. It is very likely that fertilisation is blocked because male and female gametes are unhealthy (we have data from other projects that support this). But, for the sake of argument, let's consider the sexual stages to be suitably healthy for experiments.... to study Cp23 in sexual stages we would have to perfectly time gene excision to occur only during sexual stages (never been done) and then look for random phenotypes because we do not know what we would be looking for. Male motility? Male gamete egress and movement has never been caught on live microscopy. Again, this request (if even possible within the current system) is another manuscript altogether.

We also strongly disagree with the reviewer's assertion that our data on Cp23's function is 'not conclusive'. We show very conclusively that induced KO of Cp23 affects gliding motility of the parasite, leading to a block in infection. It should also be noted that it was very challenging to perform these assays – Cryptosporidium merozoite movement has only been caught on live microscopy by a handful of researchers and this is the first time it has been done quantitatively, and to add to that complexity it was done comparatively with a transgenic strain.

One of the major points of the paper is the 50-fold increase in transfection efficiency. As the authors have indicated in their response, there is variability in transfection depending on the fresh parasite batch; therefore, it is pertinent to show the raw luminescence data for the readers to fully understand the variability between assays.

We believe this is one of many 'major points', but we appreciate that the reviewer is particularly interested in this result. We have included a graph of the non-normalised RLU for Supp Fig 1c, since the overall increase in transfection efficiency appears to be the main focus of the reviewer.

The authors have provided a repeat of their pilot CRISPR screen (Fig. S2), but the results are not consistent and appear somewhat different. For example, why does cgd1_1900 show contrasting results in terms of fitness in the repeat experiment?

In vivo genetic screens show variation because there are a number of variables that cannot be controlled. Just to name two: parasite populations (this is a passaged strain that is not clonal) and the microbiota of the mice (which is known to influence infection) are not fixed. Given these very difficult hurdles we are still able to achieve consistent results with our screens and we report our R values for each repeat. Importantly, our screen has faithfully reproduced individual KO effects on survival for all genes with this data, including thymidine kinase, dihydrofolate reductase, ribonucleotide reductase, Cp23, apical glycoprotein 1, and apical glycoprotein 2.

CP23 localization data is still unclear and requires quantification and showing additional images for readers to fully understand the changes in localization of this protein under different permeabilization conditions. The authors have still not addressed the question of why the CP23 staining in permeabilized parasites for the expanded parasites in Fig.4F appears different from the permeabilized parasite panel in Fig.4H.

Our explanation of the permeabilization data is in the manuscript (below) and in the previous reviewer response. We have now added quantification of the permeabilization data within the manuscript, which does not change our conclusions.

‘When permeabilised with Triton X-100, sporozoites demonstrated faint staining with the Cp23 antibody (Fig. 4h). Without permeabilisation, C. parvum sporozoites, surprisingly, demonstrated either a complete lack of Cp23 signal or a heightened intensity of signal throughout the parasite (Fig. 4h). In these high intensity parasites, we can detect the internal control antibody (CpTrpB - Cryptosporidium tryptophan synthase beta) at low levels. This suggests that 1) high intensity Cp23 parasites are weakly permeabilised, 2) permeabilisation with Triton X-100 affects Cp23 localisation, likely by disrupting its membrane association, and 3) Cp23 is likely not exposed on the surface of the sporozoite.’

Regarding the differences between Figure 4h and 4f: Figure 4f is expansion microscopy. With expansion microscopy, samples are permeabilised by physical separation of the gel that the proteins are attached to – there is no detergent. In the permeabilization experiment 4h, parasites are treated with Triton X-100, a detergent that is known to affect the membrane and membrane proteins. As noted above, we believe Triton X-100 affects Cp23 localisation, likely by disrupting its membrane association.